# Incidence of invasive pneumococcal disease after introduction of the 13-valent conjugate pneumococcal vaccine in British Columbia: A retrospective cohort study

**Nirma Khatri Vadlamudi**[1], **David M. Patrick**[2,3], **Linda Hoang**[3,4], **Manish Sadarangani**[5,6], **Fawziah Marra**[1,2] *

1 Faculty of Pharmaceutical Sciences, University of British Columbia, Vancouver, Canada, 2 School of Population and Public Health, Faculty of Medicine, University of British Columbia, Vancouver, Canada, 3 British Columbia Centre for Disease Control, Vancouver, Canada, 4 Department of Pathology and Laboratory Medicine, Faculty of Medicine, University of British Columbia, Vancouver, Canada, 5 Department of Pediatrics, Faculty of Medicine, University of British Columbia, Vancouver, Canada, 6 Vaccine Evaluation Center, BC Children's Hospital Research Institute, Vancouver, Canada

* fawziah@mail.ubc.ca

**Data Availability Statement:** Data cannot be shared publicly because the release of the data for my study was under the approval of the data

## Abstract

### Background

A significant reduction in invasive pneumococcal disease (IPD) has been reported, across all ages, following the implementation of 7-valent conjugate pneumococcal vaccine (PCV7) globally, as part of infant immunization programs. We explored the additional impact of PCV13 on IPD over a 14-year period.

### Methods

Using provincial laboratory surveillance and hospitalization data (N = 5791), we calculated the annual incidence of IPD following the implementation of PCV13 vaccine. Poisson regression was used to evaluate changes in the overall incidence of IPD, and serotype-specific IPD between PCV7 (2004–10) and PCV13 (2011–2015) eras.

### Results

Overall, IPD rates have seen a modest decline in the PCV13 compared to the PCV7 era (IRR 0.84; 95% CI: 0.79–0.89); this was seen in children ≤2 years of age, and the majority of the adult cohort. Rates of vaccine-type IPD (PCV7 and PCV13) also decreased in the PCV13 era. In contrast, IPD incidence related to non-PCV13 (IRR: 1.56; 95%CI:1.43–1.72) and non-vaccine serotypes (IRR: 2.12; 95%CI:1.84–2.45) increased in the PCV13 era compared to the PCV7 era.

### Conclusions

A modest reduction in IPD from the PCV13 vaccine was observed, with gains limited to the immunized cohort and adults. However, a significant increase in non-vaccine serotypes emphasizes the need for continued surveillance.

providers and governed by the Research Agreement between the data providers and the research institute. The data can only be used for the approved purposes and accessed by approved users. Data requests may be sent to PopData at dataaccess@popdata.bc.ca.

**Funding:** Invasive pneumococcal disease surveillance program is funded by British Columbia Ministry of Health. Population Data BC subsidized funding for the data that was required for NKV's doctoral thesis project. The remainder of the data acquisition costs and salary for NKV was covered by an independent investigator grant from Pfizer, Canada to Dr. Marra. The funders had no role in the data analyses and interpretation.

**Competing interests:** The authors have declared that no competing interests exist.

**Abbreviations:** BC, British Columbia; BCCDC, British Columbia Center for Disease Control and Prevention; IPD, Invasive pneumococcal disease; Non-PCV13 serotypes, Invasive pneumococcal disease from serotypes not included in the conjugate vaccines (they consist of serotypes comprised of the additional 11 serotypes included in the 23-valent polysaccharide pneumococcal vaccine and NVT serotypes); NVT, Invasive pneumococcal disease from non-vaccine *Streptococcus pneumoniae* serotypes; PCV7, 7-valent pneumococcal conjugate vaccine; PCV13, 13-valent pneumococcal conjugate vaccine; PPV23, 23-valent pneumococcal polysaccharide vaccine; VT, Vaccine type.

## Introduction

A common pathogen, *Streptococcus pneumoniae* colonizes the nasopharynx of individuals across all ages [1]. While this gram-positive bacterium is carried asymptomatically in most healthy individuals, it causes substantial morbidity and mortality in high risk populations, through a wide range of invasive (pneumococcal meningitis, pneumococcal bacteremia, and pneumococcal septicemia) and non-invasive (pneumococcal pneumonia, acute otitis media, sinusitis) pneumococcal syndromes [1]. At-risk individuals include children under 2 years of age, adults 65 years and over, individuals with underlying co-morbidities, immunosuppressive diseases, high risk behaviors (e.g. smoking, alcoholism) or environmental conditions (e.g. homelessness, crowded living conditions) [2–5].

In 2003, the Province of British Columbia (BC), Canada introduced the 7-valent pneumococcal conjugate vaccine (PCV7) to their routine, infant immunization program as a 3+1 dosing schedule at 2, 4, 6 months, followed by booster at 15–18 months [6]. The vaccine dose schedule for PCV7 was changed to a 2+1 schedule in 2007, whereby two doses were administered at 2 and 4 months of age with a booster at 12 months [6–8]; the vaccination rate of PCV7 was between 79–83% among the eligible population, between 2007 and 2009 [9]. In 2010 BC adopted PCV13, containing an additional six serotypes: 1, 3, 5, 6A, 7F, and 19A for their infant immunization program [10]; the vaccination rate of PCV13 has remained over 82% for the eligible population since 2010 [9]. BC has recommended the 23-valent pneumococcal vaccine (PPV23), with additional 11 serotypes: 2, 8, 9N, 10A, 11A, 12F, 15B, 17F, 20, 22F and 33F, universally for adults aged 65 years and older and individuals over 2 years of age with underlying co-morbidities and immunocompromised health conditions since the early 80's [11].

The early impact of PCV7 and PCV13 vaccine has been reported in many countries across the world, with an overall reduction in IPD [12–15]. Multiple regions in Canada have reported this trend [2, 12, 16], with the most prominent reduction in children younger than 5 years [2, 12]. In BC, the impact of PCV7 implementation in provincial infant immunization program in BC was previously reported [6], but the effect of PCV13 has not yet been evaluated. Further reason to conduct this study was the recent studies that have shown a possible increase in IPD rates due to an increase in non-PCV13 serotypes, particularly in countries wherein a switch to a 2+1 schedule occurred [13, 14]. This study evaluates the impact of the 13-valent pneumococcal conjugate vaccine, over a 14-year period, on children and adults in British Columbia, a Canadian province with approximately 4.8 million residents [17].

## Methods

### Ethics statement

Study protocols were approved by the University of British Columbia Institutional Review Board (IRB) as well as the BC Ministry of Health. Datasets used for this study were obtained from Population Data BC, which uses personal health numbers of patients to link the various datasets, and provides an anonymized dataset to researchers.

### Study population and duration

In 2002, British Columbia implemented a passive surveillance program, wherein all invasive *S. pneumoniae* isolates are submitted to the provincial public health laboratory for serotyping. In addition, all patients admitted to a hospital with an IPD diagnosis are recorded in the Discharge Abstract Database (DAD) through ICD9/10 codes. This retrospective cohort study utilized these administrative databases to identify all IPD cases, between January 1, 2002 to December 31, 2015, that were submitted to the provincial laboratory and/or reported in the hospitalization (DAD) database with a primary or secondary diagnosis of IPD. This cohort was then linked to a

provincial patient registry containing information regarding patient age, sex and geographic location [18]. Missing sex and location data was approximated using known proportions.

## Case definition and laboratory testing

An IPD case was defined as: a patient for whom the laboratory culture confirmed the presence of *S. pneumoniae* in cerebrospinal fluid, blood or a normally sterile site [10] and/or was hospitalized with one of the following diagnoses: pneumococcal meningitis (ICD9:320.1, ICD10: G00.1), pneumococcal bacteremia (ICD9:790.7, ICD10:R78.81) or pneumococcal septicemia (ICD9:038.9, ICD10:A40.3). Any additional laboratory data or hospitalization episode, within 30 days, was considered to be a part of the same index case. Positive blood or cerebrospinal fluid samples, wherein diagnostic codes were missing, were considered cases of pneumococcal bacteremia and pneumococcal meningitis, respectively. Isolates were tested for *S. pneumoniae* at the BCCDC using the Quellung reaction (Staten Serum Institute, Copenhagen, Denmark) and sent on to the National Microbiology Laboratory, Winnipeg, Canada for enhanced serotyping. When serotype data were missing, the serotype was approximated using vaccine and non-vaccine serotype case proportion for the specific age groups in the specific year.

The vaccine-type serotypes described in this study are those included in the PCV7 vaccine (4B, 6B, 9V, 14, 18C, 19F, 23F), the additional six serotypes included in the PCV13 vaccine ($\Delta6$ = 1, 3, 5,6A, 7F, 19A), and the additional 11 serotypes included in the PPV23 vaccine ($\Delta11$ = 2, 8, 9N, 10A, 11A, 12F, 15B, 17F, 20, 22F and 33F). All remaining serotypes, not included in any formulation, were labeled: non- vaccine-type serotypes (NVT). Non-PCV13 serotypes comprised of aforementioned additional 11 serotypes included in the PPV23 and NVT serotypes. Underlying comorbidities and immunosuppressive conditions of the IPD cases were identified through hospitalization data, and were used to calculate Charlson comorbidity score [19].

## Statistical analysis

Incidence and mortality rate calculations used population estimates from Statistics Canada and BC Statistics, as recorded in the registry [17]. For age-specific reporting, cases were grouped as follows: 0–2 years, 3–5 years, 6–17 years, 18–49 years, 50–64 years, 65–74 years, 75–84 years and 85 years and above. Annual crude, age, and vaccine serotype-specific incidence of IPD was calculated by dividing cases by population size for specific year in BC. Average annual age- and sex-specific rates were calculated across vaccine program periods: (i) pre-vaccine period or baseline (2002–2003), (ii) PCV-7 vaccination period or era (2004–2010), and (iii) PCV13 vaccination period or era (2011–2015). The Generalized linear regression (Poisson-family) models were used to assess the impact of vaccine programs by estimating rate ratios. The primary analysis was the comparison of PCV13 vaccination period (2011–2015) to PCV-7 vaccination period (2004–2010) for overall IPD rates. A number of secondary analyses were conducted including comparing overall IPD rates, vaccine-specific, non-vaccine serotypes and mortality in the PCV7 and PCV13 vaccine periods to the pre-vaccine period (2002–2003). Changes and trends in annual incidence rates were evaluated using the Cochran Armitage test, from 2002 to 2015.All statistical comparisons were assessed at a 5% significance level. Bonferroni correction was applied for multiple comparisons. Analyses were performed using SAS® 9.4 (SAS Institute Inc., Cary, NC, USA) and R v3.1.2 open source software.

## Results

### Study population and demographics

Table 1 shows the baseline characteristics of the study population. A total of 5791 invasive pneumococcal disease cases were reported in the 14-year period, of these cases, the majority

**Table 1. Characteristics of invasive pneumococcal disease cohort, British Columbia (Canada), 2002–2015.**

| | 2002–2015 | |
|---|---|---|
| | Cases[a] | % |
| **Total** | 5791 | |
| Discharge Abstract Database (DAD) | 4645 | |
| Provincial Public Health Laboratories (PHL) | 4490 | |
| Patients in both DAD and PHL databases | 3344 | |
| **Sex** | | |
| Male | 3176 | 54.8 |
| Female | 2615 | 45.2 |
| **Age groups** | | |
| 0-2y | 354 | 6.1 |
| 3-5y | 203 | 3.5 |
| 6-17y | 225 | 3.9 |
| 18–49 | 1408 | 24.3 |
| 50–64 | 1329 | 22.9 |
| 65–74 | 808 | 14.0 |
| 75–84 | 848 | 14.6 |
| 85+ | 616 | 10.6 |
| **Income Quintile** | | |
| Low (1) | 1734 | 29.9 |
| 2 | 1140 | 19.7 |
| 3 | 1026 | 17.7 |
| 4 | 985 | 17.0 |
| High(5) | 906 | 15.6 |
| **Location** | | |
| Rural | 705 | 12.2 |
| Urban | 5086 | 87.8 |
| **Clinical Diagnosis** | | |
| Pneumococcal Bacteremia | 3484 | 60.2 |
| Pneumococcal Septicemia | 2012 | 34.7 |
| Pneumococcal Meningitis | 295 | 5.1 |
| **Comorbidities and immunocompromised conditions** | | |
| None | 2378 | 41.1 |
| 0–1 | 358 | 6.2 |
| $\geq 2$ | 3055 | 52.8 |
| Top 5 conditions: | | |
| Any malignancy[b] | 1065 | |
| Mild Liver Disease | 988 | |
| Chronic Obstructive Pulmonary Disease | 911 | |
| Diabetes | 768 | |
| Congestive Heart Failure | 675 | |
| **Charlson Index Score[c]** | | |
| 0 | 1550 | 26.8 |
| 1 | 530 | 9.2 |
| 2 | 601 | 10.4 |
| 3+ | 3110 | 53.7 |

[a] repeat isolate or hospitalization with IPD within 30 days was considered part of same case.

[b] Any malignancy, including lymphoma and leukemia, except malignant neoplasm of skin.

[c] Calculated based on age, comorbidities and immunocompromised conditions.

were males (54.8%). Among children, the majority were aged ≤ 2 years (6.1%) with a median age of 3 years (interquartile range [IQR], 1–6 years). For adults, defined as those 18 years of age and older, the majority were 18–49 years (24.3%) with a median age of 62 years (IQR], 47-77years). The most common clinical diagnosis was pneumococcal bacteremia (60.2%), followed by pneumococcal septicemia (34.7%), and pneumococcal meningitis (5.1%) with most cases occurring in individuals between the ages of 18 and 64 years and those with comorbid and/or immunocompromising conditions (eTable 1 in S1 Appendix).

A seasonal pattern in the IPD incidence was observed, with 1775 cases (30.7%) occurring during January–March, compared with 862 cases (14.9%) during July–September (eFigure 1 in S1 Appendix). Of the 5791 cases in the 14-year study period, a total 4490 (77.5%) non-duplicate or non-repeat invasive isolates of *S.pneumoniae* were reported; seven serotypes accounted for 45% of all cases: serotype 3 (396 isolates, 9%), 19A (392, 9%), 22F (372, 8%), 5 (351,8%), 7F (302,7%) and 14 (204, 5%).

## Incidence of IPD

An increasing trend was observed in the overall IPD incidence from 8.3 per 100,000 in 2002 to 8.7 per 100,000 in 2015 (Fig 1), although the temporal trend evaluation indicated no significant change (p-value = 0.21). The highest incidence was in 2007, due to a serotype 5 epidemic among indigent people, while the lowest was 2004, directly following the introduction of PCV7 in BC. IPD due to PCV7 serotypes decreased from 3.9 per 100,000 (2002) to 0.9 per 100,000 (2015) while IPD related to PCV13 serotypes increased from 0.8 per 100,000 (2002) to

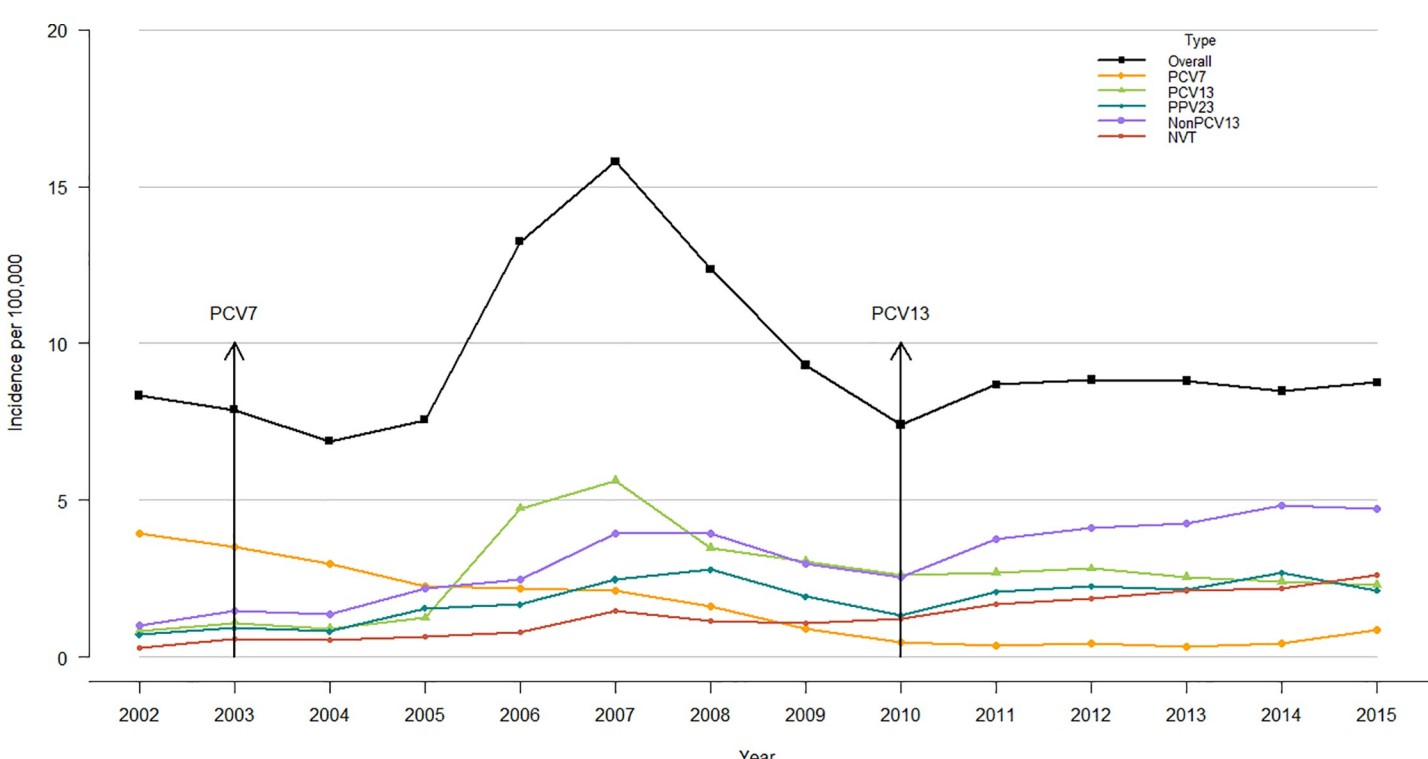

**Fig 1. Invasive pneumococcal disease incidence from 2002 to 2015.** Overall = laboratory and hospitalization data. PCV7 = serotypes in the 7- valent pneumococcal conjugate vaccine. PCV13 = additional six serotypes in the PCV13 vaccines not in PCV7 vaccine serotype. PPV23 = additional 11 serotypes not in the PCV13 vaccine. NVT = non-vaccine serotype. Non-PCV13 = serotypes not included in the conjugate vaccines (they consist of serotypes comprised of the additional 11 serotypes included in the 23-valent polysaccharide pneumococcal vaccine and NVT serotypes). While no changes were noted for overall IPD (p = 0.2138), significant changes occurred for PCV7, PCV13, PPV23, NVT, Non-PCV13 serotype IPD (p<0.0001).

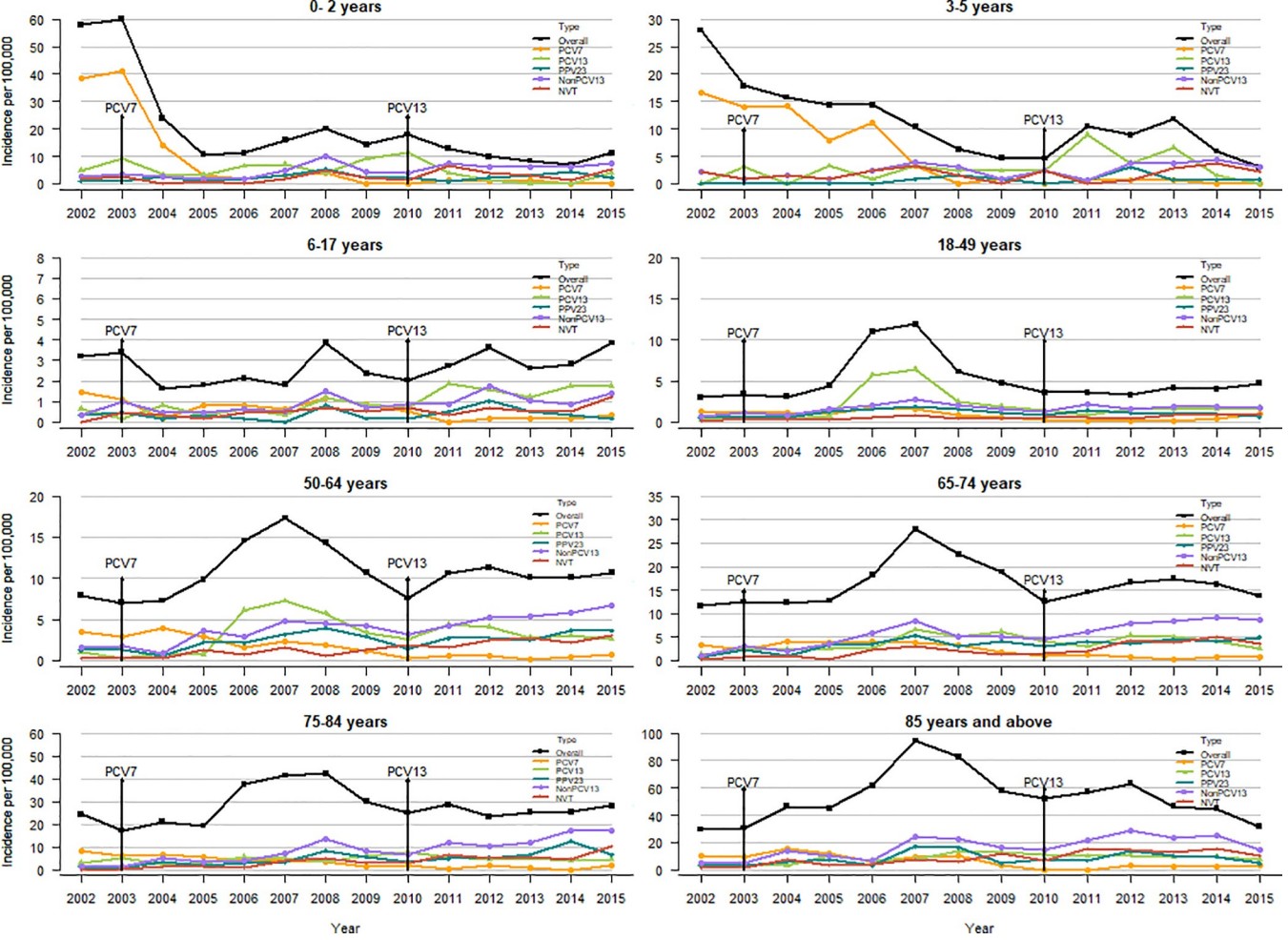

**Fig 2. Invasive pneumococcal disease incidence from 2002 to 2015 by age.** Overall = laboratory and hospitalization data. PCV7 = serotypes in the 7- valent pneumococcal conjugate vaccine. PCV13 = additional six serotypes in the PCV13 vaccines not in PCV7 vaccine serotype. PPV23 = additional 11 serotypes not in the PCV13 vaccine. NVT = non-vaccine serotype. Non-PCV13 = serotypes not included in the conjugate vaccines (they consist of serotypes comprised of the additional 11 serotypes included in the 23-valent polysaccharide pneumococcal vaccine and NVT serotypes).

2.3 per 100,000 (2015). However, a significant increase in non-PCV13 serotypes was observed between 2002 and 2015, changing from 1.0 per 100,000 to 4.73 per 100,000 (Fig 1).

When evaluating the trends by age we saw that IPD incidence declined by 81% in children aged ≤2 years and 89% in 3–5 years old, but the incidence has increased in all other age groups: 20% in 6–17 years, 56% in 18–49 years, 35% in 50–64 years, 19% in 65–74 years, 15% in 75–84 years and 7% in ≥ 85 years (Fig 2).

## Overall IPD incidence

Table 2 shows the total number of cases within each immunization period of our study. When comparing IPD incidence between the two vaccine eras, a modest reduction of 16% was observed in the PCV13 era compared with the PCV7 era (IRR 0.84; 95% CI: 0.79–0.89). Although overall reduction in IPD rates was only significant among ≤ 2 years (IRR:0.60; 95% CI: 0.45–0.80) in the children category, the majority of the adults had a significant reduction in overall IPD rates: 18–49 years (IRR 0.62; 95% CI: 0.55–0.70), 75–84 years (IRR 0.84; 95% CI: 0.73–0.97), and those 85 years and over (IRR 0.76; 95% CI: 0.64–0.89). When comparing the

**Table 2. Number of cases and incidence of invasive pneumococcal disease after introduction of pneumococcal conjugate vaccines in BC compared with the baseline average of 2002–2003, by all age group and serotype.**

| | Pre-vaccine | | PCV7 | | PCV13 | | Incidence Rate Ratios (95% CI) | | |
|---|---|---|---|---|---|---|---|---|---|
| | 2002–2003 | | 2004–2010 | | 2011–2015 | | PCV7/ | PCV13/ | PCV13/ |
| | cases | IR | cases | IR | cases | IR | Pre-vaccine | Pre-vaccine | PCV7 |
| **All age groups** | | | | | | | | | |
| Overall IPD | 667 | 8.11 | 3122 | 10.37 | 2002 | 8.71 | 1.28(1.18–1.39) | 1.07(0.98–1.17) | 0.84(0.79–0.89) |
| PCV7 | 306 | 3.72 | 526 | 1.75 | 112 | 0.49 | 0.47(0.41–0.54) | 0.13(0.11–0.16) | 0.28(0.23–0.34) |
| PCV13 | 79 | 0.96 | 947 | 3.15 | 588 | 2.56 | 3.27(2.62–4.15) | 2.66(2.12–3.39) | 0.81(0.73–0.9) |
| PPV23 | 67 | 0.81 | 537 | 1.78 | 515 | 2.24 | 2.19(1.71–2.85) | 2.75(2.15–3.58) | 1.26(1.11–1.42) |
| NVT | 35 | 0.43 | 297 | 0.99 | 481 | 2.09 | 2.32(1.66–3.35) | 4.92(3.55–7.06) | 2.12(1.84–2.45) |
| Non-PCV13 | 102 | 1.24 | 834 | 2.77 | 996 | 4.33 | 2.23(1.83–2.76) | 3.5(2.87–4.31) | 1.56(1.43–1.72) |
| **Age ≤2 years** | | | | | | | | | |
| All | 144 | 58.98 | 145 | 16.28 | 65 | 9.81 | 0.28(0.22–0.35) | 0.17(0.12–0.22) | 0.60(0.45–0.80) |
| PCV7 | 97 | 39.73 | 32 | 3.59 | 5 | 0.75 | 0.09 (0.06–0.13) | 0.02 (0.01–0.04) | 0.21 (0.07–0.49) |
| PCV13 | 17 | 6.96 | 57 | 6.40 | 12 | 1.81 | 0.92(0.55–1.63) | 0.26(0.12–0.54) | 0.28(0.14–0.51) |
| PPV23 | * | 0.82 | 23 | 2.58 | 17 | 2.57 | 3.15(0.93–19.65) | 3.13(0.9–19.75) | 0.99(0.52–1.85) |
| NVT | 5 | 2.05 | 14 | 1.57 | 27 | 4.07 | 0.77(0.29–2.38) | 1.99(0.83–5.87) | 2.59(1.38–5.08) |
| Non-PCV13 | 7 | 2.87 | 37 | 4.16 | 44 | 6.64 | 1.45(0.69–3.55) | 2.32(1.11–5.63) | 1.6(1.03–2.49) |
| **Age 3–5 years** | | | | | | | | | |
| All | 60 | 22.96 | 89 | 10.00 | 54 | 7.99 | 0.44(0.31–0.61) | 0.35(0.24–0.5) | 0.8(0.57–1.12) |
| PCV7 | 40 | 15.30 | 47 | 5.28 | * | 0.44 | 0.35(0.23–0.53) | 0.03 (0.01–0.08) | 0.08(0.02–0.23) |
| PCV13 | * | 1.53 | 18 | 2.02 | 28 | 4.15 | 1.32(0.49–4.58) | 2.71(1.06–9.15) | 2.05(1.14–3.77) |
| PPV23 | - | … | * | 0.45 | 8 | 1.18 | … | … | 2.63(0.83–9.87) |
| NVT | * | 1.53 | 15 | 1.69 | 13 | 1.92 | 1.1(0.4–3.86) | 1.26(0.45–4.46) | 1.14(0.54–2.4) |
| Non-PCV13 | * | 1.53 | 19 | 2.13 | 21 | 3.11 | 1.4(0.52–4.81) | 2.03(0.77–6.96) | 1.46(0.78–2.73) |
| **Age 6–17 years** | | | | | | | | | |
| All | 41 | 3.30 | 94 | 2.23 | 90 | 3.13 | 0.68(0.47–0.99) | 0.95(0.66–1.39) | 1.4(1.05–1.87) |
| PCV7 | 16 | 1.29 | 29 | 0.69 | 5 | 0.17 | 0.54(0.29–1.01) | 0.14(0.04–0.35) | 0.25(0.09–0.6) |
| PCV13 | 5 | 0.40 | 30 | 0.71 | 47 | 1.64 | 1.77(0.75–5.2) | 4.07(1.78–11.73) | 2.3(1.46–3.67) |
| PPV23 | 5 | 0.40 | 11 | 0.26 | 15 | 0.52 | 0.65(0.24–2.06) | 1.3(0.5–3.99) | 2(0.92–4.46) |
| NVT | * | 0.24 | 20 | 0.48 | 19 | 0.66 | 1.97(0.68–8.36) | 2.74(0.93–11.67) | 1.39(0.74–2.62) |
| Non-PCV13 | 8 | 0.64 | 31 | 0.74 | 34 | 1.18 | 1.14(0.55–2.67) | 1.84(0.9–4.28) | 1.61(0.99–2.63) |
| **Age 18–49 years** | | | | | | | | | |
| All | 125 | 3.19 | 884 | 6.39 | 399 | 3.96 | 2.01(1.67–2.43) | 1.24(1.02–1.52) | 0.62(0.55–0.7) |
| PCV7 | 50 | 1.28 | 141 | 1.02 | 39 | 0.39 | 0.8(0.58–1.11) | 0.3(0.2–0.46) | 0.38(0.26–0.54) |
| PCV13 | 13 | 0.33 | 380 | 2.75 | 148 | 1.47 | 8.29(4.98–15.19) | 4.43(2.62–8.22) | 0.53(0.44–0.64) |
| PPV23 | 23 | 0.59 | 173 | 1.25 | 107 | 1.06 | 2.13(1.41–3.38) | 1.81(1.18–2.91) | 0.85(0.67–1.08) |
| NVT | 12 | 0.31 | 68 | 0.49 | 77 | 0.76 | 1.61(0.9–3.12) | 2.5(1.42–4.83) | 1.55(1.12–2.16) |
| Non-PCV13 | 35 | 0.89 | 241 | 1.74 | 184 | 1.83 | 1.95(1.39–2.83) | 2.05(1.45–2.99) | 1.05(0.86–1.27) |
| **Age 50–64 years** | | | | | | | | | |
| All | 108 | 7.46 | 700 | 11.67 | 521 | 10.56 | 1.57(1.28–1.93) | 1.42(1.16–1.75) | 0.9(0.81–1.01) |
| PCV7 | 46 | 3.18 | 114 | 1.90 | 23 | 0.47 | 0.6(0.43–0.85) | 0.15(0.09–0.24) | 0.25(0.15–0.38) |
| PCV13 | 9 | 0.62 | 235 | 3.92 | 168 | 3.40 | 6.3(3.44–13.25) | 5.48(2.97–11.56) | 0.87(0.71–1.06) |
| PPV23 | 19 | 1.31 | 143 | 2.38 | 151 | 3.06 | 1.82(1.16–3.03) | 2.33(1.49–3.88) | 1.28(1.02–1.61) |
| NVT | 5 | 0.35 | 67 | 1.12 | 120 | 2.43 | 3.24(1.44–9.24) | 7.04(3.2–19.9) | 2.18(1.62–2.95) |
| Non-PCV13 | 24 | 1.66 | 210 | 3.50 | 271 | 5.49 | 2.11(1.42–3.31) | 3.31(2.23–5.16) | 1.57(1.31–1.88) |
| **Age 65–74 years** | | | | | | | | | |
| All | 71 | 12.03 | 408 | 17.92 | 329 | 15.71 | 1.49(1.17–1.93) | 1.31(1.02–1.7) | 0.88(0.76–1.01) |

*(Continued)*

**Table 2.** (Continued)

| | Pre-vaccine | | PCV7 | | PCV13 | | Incidence Rate Ratios (95% CI) | | |
|---|---|---|---|---|---|---|---|---|---|
| | 2002–2003 | | 2004–2010 | | 2011–2015 | | PCV7/ | PCV13/ | PCV13/ |
| | cases | IR | cases | IR | cases | IR | Pre-vaccine | Pre-vaccine | PCV7 |
| **All age groups** | | | | | | | | | |
| PCV7 | 17 | 2.88 | 67 | 2.94 | 16 | 0.76 | 1.02(0.61–1.8) | 0.27(0.13–0.53) | 0.26(0.15–0.44) |
| PCV13 | 11 | 1.86 | 100 | 4.39 | 83 | 3.96 | 2.36(1.32–4.66) | 2.13(1.19–4.22) | 0.9(0.67–1.21) |
| PPV23 | 9 | 1.53 | 76 | 3.34 | 89 | 4.25 | 2.19(1.16–4.69) | 2.79(1.49–5.95) | 1.27(0.94–1.73) |
| NVT | * | 0.51 | 37 | 1.62 | 81 | 3.87 | 3.2(1.16–13.24) | 7.61(2.85–31.01) | 2.38(1.63–3.55) |
| Non-PCV13 | 12 | 2.03 | 113 | 4.96 | 170 | 8.12 | 2.44(1.4–4.67) | 3.99(2.32–7.57) | 1.64(1.29–2.08) |
| **Age 75–84 years** | | | | | | | | | |
| All | 81 | 20.65 | 461 | 31.08 | 306 | 26.15 | 1.5(1.2–1.92) | 1.27(1–1.63) | 0.84(0.73–0.97) |
| PCV7 | 28 | 7.14 | 56 | 3.78 | 11 | 0.94 | 0.53(0.34–0.84) | 0.13(0.06–0.26) | 0.25(0.12–0.46) |
| PCV13 | 16 | 4.08 | 72 | 4.85 | 56 | 4.79 | 1.19(0.71–2.12) | 1.17(0.69–2.11) | 0.99(0.69–1.4) |
| PPV23 | 5 | 1.27 | 62 | 4.18 | 84 | 7.18 | 3.28(1.46–9.38) | 5.63(2.53–16) | 1.72(1.24–2.39) |
| NVT | * | 0.25 | 39 | 2.63 | 76 | 6.49 | 10.31(2.24–183.01) | 25.47(5.66–449.4) | 2.47(1.69–3.67) |
| Non-PCV13 | 6 | 1.53 | 101 | 6.81 | 160 | 13.6727 | 4.45(2.13–11.4) | 8.94(4.33–22.74) | 2.01(1.57–2.58) |
| **Age ≥85 years** | | | | | | | | | |
| All | 37 | 29.94 | 341 | 63.43 | 238 | 48.00 | 2.12(1.53–3.02) | 1.6(1.15–2.3) | 0.76(0.64–0.89) |
| PCV7 | 12 | 9.71 | 40 | 7.44 | 10 | 2.02 | 0.77(0.41–1.53) | 0.21(0.09–0.48) | 0.27(0.13–0.52) |
| PCV13 | * | 3.24 | 55 | 10.23 | 46 | 9.28 | 3.16(1.3–10.44) | 2.87(1.17–9.51) | 0.91(0.61–1.34) |
| PPV23 | * | 3.24 | 45 | 8.37 | 44 | 8.87 | 2.59(1.05–8.59) | 2.74(1.11–9.11) | 1.06(0.7–1.61) |
| NVT | * | 1.62 | 37 | 6.88 | 68 | 13.71 | 4.25(1.3–26.16) | 8.47(2.66–51.6) | 1.99(1.34–3) |
| Non-PCV13 | 6 | 4.85 | 82 | 15.25 | 112 | 22.5867 | 3.14(1.49–8.08) | 4.65(2.23–11.9) | 1.48(1.12–1.97) |

IR = Incidence rate per 100000. IRR = incidence rate ratio. PCV = pneumococcal conjugate vaccine. PCV7 = serotypes in the pneumococcal conjugate vaccine 7.

PCV13 = additional six serotypes in the PCV13 vaccines not in PCV7 vaccine. PPV23 = additional 11 serotypes not in the PCV13 vaccine. NVT = non-vaccine serotype.

Non-PCV13 = any serotype not included in the PCV13 vaccine.

* = cases less than 5. - = no cases. ⋯ = cannot compute.

two vaccine periods to baseline, a statistically significant increase of 28% in overall IPD incidence was seen in the PCV7 era (IRR 1.28; 95% CI: 1.18–1.39), but no significant change was seen in the PCV13 era (IRR 1.07; 95% CI: 0.98–1.17); the majority of this increase was seen in the adult population (Table 2).

## IPD incidence trends from PCV7 serotypes

Following the introduction of PCV7, IPD incidence for PCV7 serotypes declined significantly, and after switching to PCV13, IPD incidence for PCV7 serotypes continued to decline (IRR 0.28; 95% CI: 0.23–0.34) in all age groups when compared to the PCV7 period (Table 2). These reductions were due to decreases in serotypes 4, 14, 18C, 19F and 23F in the PCV7 era; significant decreases in serotypes 6B and 9V were only seen after introduction of PCV13 (Table 3).

## IPD incidence trends from the additional PCV13 serotypes

As seen in Table 2, following PCV13 introduction, IPD rates for the 6 additional PCV13 serotypes (or Δ6) declined by 19% when compared to PCV7 era (IRR 0.81; 95% CI: 0.73–0.90). However, this decrease was only significant in children ≤ 2 years of age (IRR 0.28; 95% CI: 0.14–0.51) and adults between the ages of 18–49 years of age (IRR 0.53; 95% CI: 0.44–0.64). In contrast, significant increases in PCV13-type IPD were noted in ages 3–5 (IRR 2.05; 95% CI:

**Table 3. Serotype cases and incidence of invasive pneumococcal disease after introduction of pneumococcal conjugate vaccines in BC compared with the baseline average of 2002–2003.**

| | Pre-vaccine | | PCV7 | | PCV13 | | Incidence Rate Ratios (95% CI) | | |
|---|---|---|---|---|---|---|---|---|---|
| | 2002–2003 | | 2004–2010 | | 2011–2015 | | PCV7/ | PCV13/ | PCV13/ |
| | cases | IR | cases | IR | cases | IR | Pre-vaccine | Pre-vaccine | PCV7 |
| **PCV7 serotypes** | | | | | | | | | |
| 4 | 40 | 0.49 | 98 | 0.33 | 51 | 0.19 | 0.67(0.47–0.98) | 0.38(0.25–0.58) | 0.57(0.4–0.8) |
| 6B | 30 | 0.36 | 78 | 0.26 | 9 | 0.03 | 0.71(0.47–1.10) | 0.09(0.04–0.18) | 0.13(0.06–0.24) |
| 9V | 37 | 0.45 | 94 | 0.31 | * | 0.01 | 0.69(0.48–1.03) | 0.02(0.01–0.07) | 0.04(0.01–0.09) |
| 14 | 91 | 1.11 | 104 | 0.35 | 9 | 0.03 | 0.31(0.24–0.41) | 0.03(0.01–0.06) | 0.09(0.04–0.18) |
| 18C | 32 | 0.39 | 51 | 0.17 | 7 | 0.03 | 0.44(0.28–0.68) | 0.07(0.03–0.14) | 0.15(0.06–0.31) |
| 19F | 41 | 0.50 | 61 | 0.20 | 25 | 0.09 | 0.41(0.27–0.61) | 0.18(0.11–0.3) | 0.45(0.28–0.71) |
| 23F | 23 | 0.28 | 40 | 0.13 | 8 | 0.03 | 0.48(0.29–0.81) | 0.1(0.04–0.22) | 0.22(0.1–0.44) |
| **Additional PCV13 serotypes** | | | | | | | | | |
| 1 | * | 0.04 | 24 | 0.08 | 6 | 0.02 | 2.19(0.76–9.2) | 0.6(0.16–2.84) | 0.27(0.1–0.63) |
| 3 | 37 | 0.45 | 204 | 0.68 | 155 | 0.57 | 1.51(1.08–2.17) | 1.26(0.89–1.83) | 0.84(0.68–1.03) |
| 5 | - | ⋯ | 339 | 1.12 | 12 | 0.04 | ⋯ | ⋯ | 0.04(0.02–0.07) |
| 6A | 21 | 0.26 | 74 | 0.25 | 5 | 0.02 | 0.96(0.6–1.6) | 0.07(0.02–0.18) | 0.07(0.03–0.17) |
| 7F | 10 | 0.12 | 106 | 0.35 | 186 | 0.68 | 2.9(1.59–5.91) | 5.57(3.12–11.27) | 1.93(1.52–2.45) |
| 19A | 8 | 0.10 | 162 | 0.54 | 222 | 0.81 | 5.53(2.91–12.26) | 8.32(4.4–18.38) | 1.5(1.23–1.84) |
| **Non- PCV13 vaccine serotypes** | | | | | | | | | |
| 6C | - | ⋯ | 12 | 0.04 | 67 | 0.24 | ⋯ | ⋯ | 6.13(3.44–11.9) |
| 8 | 6 | 0.07 | 57 | 0.19 | 62 | 0.23 | 2.59(1.21–6.73) | 3.1(1.46–8.02) | 1.19(0.83–1.71) |
| 9N | 11 | 0.13 | 52 | 0.17 | 62 | 0.23 | 1.29(0.7–2.61) | 1.69(0.93–3.39) | 1.31(0.91–1.9) |
| 10A | * | 0.01 | 24 | 0.08 | 26 | 0.09 | 6.56(1.39–117.2) | 7.79(1.66–139.11) | 1.19(0.68–2.08) |
| 11A | 7 | 0.09 | 45 | 0.15 | 58 | 0.21 | 1.76(0.85–4.26) | 2.48(1.22–5.97) | 1.41(0.96–2.1) |
| 12F | 9 | 0.11 | 87 | 0.29 | 18 | 0.07 | 2.64(1.41–5.64) | 0.6(0.28–1.4) | 0.23(0.13–0.37) |
| 15A | * | 0.04 | 24 | 0.08 | 50 | 0.18 | 2.19(0.76–9.2) | 4.99(1.84–20.54) | 2.29(1.42–3.78) |
| 15B | * | 0.04 | 19 | 0.06 | 20 | 0.07 | 1.73(0.59–7.36) | 2(0.69–8.48) | 1.15(0.61–2.18) |
| 15C | * | 0.01 | 20 | 0.07 | 23 | 0.08 | 5.46(1.14–98.04) | 6.89(1.45–123.34) | 1.26(0.69–2.32) |
| 16F | 6 | 0.07 | 24 | 0.08 | 25 | 0.09 | 1.09(0.48–2.95) | 1.25(0.55–3.36) | 1.14(0.65–2.01) |
| 17F | * | 0.01 | 18 | 0.06 | 22 | 0.08 | 4.92(1.02–88.46) | 6.59(1.39–118.08) | 1.34(0.72–2.53) |
| 20 | * | 0.01 | 38 | 0.13 | 26 | 0.09 | 10.38(2.25–184.26) | 7.79(1.6–6 139.11) | 0.75(0.45–1.23) |
| 22F | 21 | 0.26 | 155 | 0.51 | 196 | 0.71 | 2.02(1.31–3.27) | 2.8(1.83–4.52) | 1.39(1.12–1.71) |
| 23A | * | 0.02 | 35 | 0.12 | 79 | 0.29 | 4.78(1.46–29.44) | 11.84(3.7–4 71.96) | 2.48(1.68–3.73) |
| 23B | * | 0.01 | 15 | 0.05 | 61 | 0.22 | 4.1(0.83–74.09) | 18.28(4–4 323.04) | 4.46(2.61–8.14) |
| 33F | * | 0.01 | 25 | 0.08 | 26 | 0.09 | 6.83(1.45–121.99) | 7.79(1.66–139.11) | 1.14(0.66–1.98) |
| 35B | * | 0.04 | 20 | 0.07 | 23 | 0.08 | 1.82(0.62–7.73) | 2.3(0.8–9.69) | 1.26(0.69–2.32) |
| 35F | * | 0.04 | 19 | 0.06 | 38 | 0.14 | 1.73(0.59–7.36) | 3.8(1.38–15.71) | 2.19(1.28–3.89) |
| 38 | * | 0.01 | 20 | 0.07 | 13 | 0.05 | 5.46(1.14–98.04) | 3.9(0.78–70.78) | 0.71(0.35–1.42) |

IR = Incidence rate per 100000. IRR = incidence rate ratio. PCV7 = serotypes in the 7- valent pneumococcal conjugate vaccine. PCV13 = additional six serotypes in the PCV13 vaccines not in PCV7 vaccine serotype.

* = cases less than 5. - = no cases. ⋯ = cannot compute.

1.14–3.77), and 6–17 years (IRR 2.30; 95% CI: 1.46–3.67). Following the PCV13 introduction, IPD from serotypes 1, 5,6A reduced significantly, whereas serotypes 7F and 19A continued to remain high; the change in serotype 3 IPD was not significant (Table 3). A year-on-year analysis of serotypes, by age groups, shows a significant decrease in serotypes 19A and 7F in children less than 18 years of age (eFigure 2 in S1 Appendix). We observed a slow decline in IPD

from 19A in adults as well during the PCV13 era, but in contrast IPD from 7F increased year-on-year in those 18 years of age and over.

## IPD incidence trends from the additional PPV23 serotypes

The incidence of IPD related to the additional 11 serotypes in PPV23 vaccine increased after the introduction of the PCV7 vaccine (IRR 2.19; 95% CI: 1.71–2.85) and further increased in the PCV13 era (IRR 1.26; 95% CI: 1.11–1.42) (Table 2); this trend was seen in all age categories in both children and adults, and was driven by PPV23 serotypes such as 8, 10A,11A, 12F,17F,20, 22F and 33F..

## IPD incidence trends from non-vaccine serotypes (NVT) and non-PCV13 serotypes

Invasive pneumococcal disease as a result of NVT has also increased significantly in the post-PCV7 period (IRR 2.32; 95% CI: 1.66–3.35), with further increases in the post-PCV13 period (IRR 4.92; 95% CI: 3.55–7.06). When comparing to pre-vaccine era, children aged ≤17 years did not experience significant change in IPD from NVT with the PCV7 introduction, however after switching to PCV13 immunization program, there was a notable increase in NVT IPD (IRR 2.04; 95% CI: 1.14–3.99) in this population. By and large, significant increases in NVT IPD has been observed in all adults in the PCV7 era (IRR: 2.89; 95%CI: 1.93–4.56) and the PCV13 era (IRR: 6.33; 95CI% 4.26–9.91).

We observed a total of 64 non-conjugate vaccine serotypes and, as outlined in Fig 3, the most common serotype was 22F (8.3%). Approximately 3% of the serotypes were: 9N (2.8%), 8

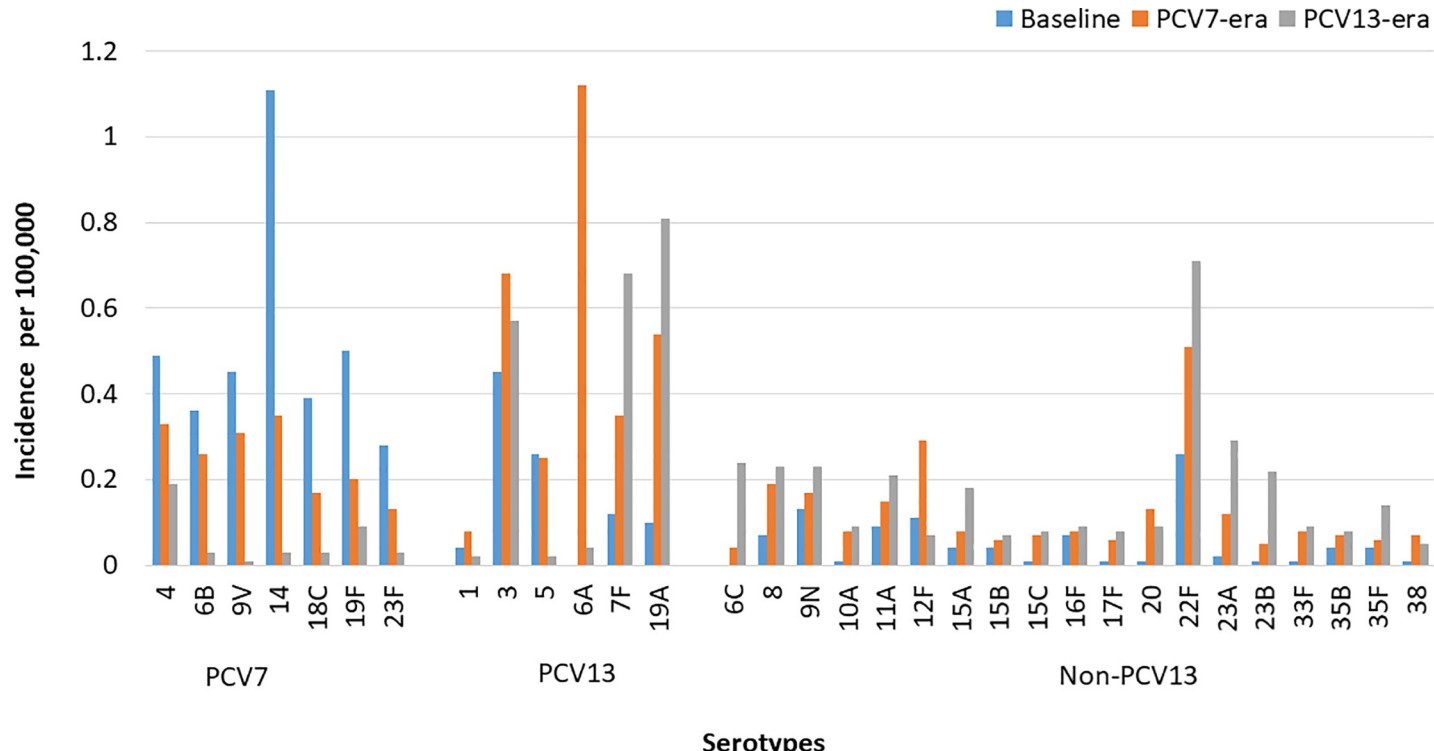

**Fig 3. Incidence of invasive pneumococcal disease by serotypes in British Columbia from 2002–2015.** PCV7 = serotypes in the 7- valent pneumococcal conjugate vaccine. PCV13 = additional six serotypes in the PCV13 vaccines not in PCV7 vaccine serotype. Non-PCV13 = serotypes not included in the conjugate vaccines (they consist of serotypes comprised of the additional 11 serotypes included in the 23-valent polysaccharide pneumococcal vaccine and NVT serotypes).

(2.8%), 23A (2.6%), 12F (2.5%), 11A (2.4%), while less than 2% were 6C (1.8%), 15A (1.7%), 23B (1.7%), 20 (1.4%). As seen in Table 2, after the introduction of the PCV13, we saw a further increase in the rates of IPD related to non-PCV13 serotypes (IRR 1.56; 1.43–1.72), mostly seen in all adult groups.

While many non-conjugate vaccine serotypes (i.e., 8, 10A, 11A, 12F, 15A, 15C, 17F, 20, 22F, 23A, 23B, 33F and 35F) have increased significantly in the PCV13 era compared to pre-vaccine era, only a handful of serotypes (6C, 15A, 22F, 23A, 23B, and 35F) have significantly increased in the PCV13 era compared to the PCV7 era (Table 3).

### Mortality rates following IPD

A decreasing trend was observed in 30-day all-cause mortality rates between 2002 (0.83 per 100,000) and 2015 (0.66 per 100,000) following IPD (eFigure3 in S1 Appendix). When evaluating the effects in the various vaccine periods, a significant reduction of 27% was observed in PCV13 era (RR: 0.73; 95% CI: 0.62–0.85) compared to the PCV7 era (eFigure3 in S1 Appendix). Similar results were seen with respect to the one-year all-cause mortality following IPD, which decreased by 23% in PCV13 era compared with PCV7 era (RR: 0.77; 95% CI: 0.68–0.88) (eFigure4 in S1 Appendix).

## Discussion

This study showed that introduction of PCV7 in the children's immunization program resulted in the near elimination of PCV7-serotype IPD in all ages compared to the pre-vaccine era. Addition of the PCV13 vaccine resulted further declines in overall IPD, in particular substantial declines in PCV13-serotype IPD were observed in the target population of children aged ≤2 years, and herd immunity was seen in adults over the age of 65 years. Furthermore, we identified substantial reductions in mortality, when compared to the PCV7 era.

Like many studies before ours, we found a substantial reduction in IPD rates in children following introduction of PCV7 and PCV13 vaccines [12–15, 20]. Declining IPD rates in adults following the use of conjugate vaccines, particularly PCV7, in children's immunization program has also been well documented [13, 15, 20–26]. Like previous studies, we observed significant declines in IPD rates within the adult population, including those aged 18–49 years as well as those above 75 years of age [15, 20]. Of note, we observed that the decline in vaccine serotype-specific IPD rates in adults were comparable across both the PCV7 era and the PCV13 era. For instance, among adults, IPD from the additional 6 serotypes of PCV13 vaccine declined by 24% in the PCV13 period when compared to PCV7 period, similarly IPD related to PCV7 serotypes reduced by 27% in the PCV7 period compared to the pre-vaccine era. Likewise, comparable declines in serotype-specific IPD in adults, in both vaccine periods, were reported in the United States [15].

While studies from Australia, the United States and the United Kingdom report substantial reductions in PCV13 VT IPD in individuals aged 65 years or older [13, 15, 20, 26], in contrast, we could not corroborate this finding. Instead we observed a persistent burden of PCV13 VT IPD in this population similar to another Canadian province, and a study conducted in Israel [21, 27]. Additionally, a prospective and population based surveillance Canadian study conducted in the province of Alberta reports lingering PCV13 VT IPD burden in those aged 65 years and older with co-morbidities and immunocompromised status [2]. Currently, neither the Canadian National Advisory Committee on Immunization (NACI) nor the United States Advisory Committee on Immunization Practice (ACIP) recommend PCV13 for the individuals aged 65 years and older [28, 29], with ACIP recently changing its recommendation from their original recommendation, made in 2014, whereby healthy adults aged ≥ 65 years were

suggested to receive this vaccine [29]. As PCV13-VT IPD in the United States has reduced to historically low levels as a result of the PCV13 childhood immunization program, but PPV23-VT IPD burden persists, they have changed their recommendation to PPV23 only for this population [29]. Given the disparity in findings reported by our study, and those listed above, it would be important to study the long-term effects of the pediatric immunization program with PCV13 and the indirect benefits it confers to this population.

We observed high variability in the effect of the PCV13 immunization program on the IPD rates related to specific serotypes, however this may be related to the reduced sample size. Similar to many large observational studies, we observed a reduction in PCV13 VT IPD for serotypes 1, 5 and 6A significantly in all ages [2, 13, 15, 21, 27, 30]. We did not see a significant change in serotype 3 disease, which corroborates findings from other studies [15, 20, 24, 31, 32], however, it should be noted that several recent studies have shown a significant increase in IPD from serotype 3 [30, 33–35]. For example, a recent study from Ontario, Canada that examined the shifting pneumococcal serotypes after introduction of PCV13 found it had no impact on incidence due to serotype 3, which continued to rise in their study population [34]. This trend is thought to be related to the lower efficacy of PCV13 against serotype 3 compared to other serotypes [36, 37] and, therefore, it is important that we continue to monitor the proportion of cases due to this serotype in both vaccinees and non-vaccinated individuals.

In this study, a significant increase was observed in PCV13 serotypes, 7F and 19A among all ages during PCV7 period. Similarly, Moore and colleagues reported substantial invasive pneumococcal disease burden from PCV13 serotypes 7F and 19A in all ages during PCV7 period [15]. In another study, Grau et al described significant increases in PCV13 serotypes 7F and 19A during PCV7 period in adults [32]. Like our study, where we saw a slow but steady decline, many have reported control of 19A after introduction of PCV13 vaccine [16, 20, 26, 31–34, 38]. But the evidence behind a substantial decline in 7F is not as clear, with the majority studies reporting declines in cases with this serotype [13, 15, 16, 21, 27, 30, 34, 39, 40], whilst several studies, like ours, report an increase [2, 12, 20, 31–33, 41]. For example, a German study reporting on pneumococcal meningitis long term trends, from 20 year period, found proportion of isolates of PCV13 serotype 7F and 19A continued to increase in adults following the PCV13 introduction [31]. Once again, given that these reports differ based on geographic locations and age, it is important to monitor and study the complex dynamics and genetic variability of the serotypes.

We found a significant decline in 30-day mortality following PCV13, this decline can be attributed to reductions in vaccine-serotype disease over time, which cause severe morbidity and mortality [42]. Likewise, many other observational studies have reported declines in 30-day mortality in the PCV13 era compared to the PCV7 era [22, 32, 43, 44]. Interestingly, while Moore et al report significant reductions in 30-day mortality in those under 65 years of age, they found an increase in case fatality rates following the introduction of PCV13 for those aged 65 years and older, especially in the presence of underlying comorbidities [15]. Given immunocompromised status puts individuals at additional risk for IPD, they are likely to benefit from PCV13 vaccination. Over time, it will be important to study real world data to understand vaccine effectiveness of PCV13 in this population.

Serotype replacement phenomenon has been reported since PCV7 introduction [45]. In this study, like other global studies, we found significant increases in the non-PCV13 serotypes IPD in most age groups during both vaccine eras [13, 21, 22]. This significant increase in non-PCV13 serotypes after PCV13 introduction has been confirmed by few systematic review and meta-analyses [23, 25, 46]. Due to the increase in non-PCV13 serotypes IPD, we only saw a modest yet significant reduction in overall IPD like other jurisdictions [27]. Therefore, it is

important to monitor changes in the serotypes over time, in order to take timely and effective measures.

However, the benefit conferred by the introduction of pneumococcal conjugate vaccines to children's immunization programs, and subsequent reductions in children, have been affected by an increase in non-vaccine-type serotype related IPD.

This study benefits from several key strengths. Firstly, our study duration includes 14 years of the provincial laboratory surveillance data, which is able to capture long-term trends in *Streptococcus pneumoniae* serotype dynamics and epidemiology over time for both the PCV7 period (7 years) and the PCV13 period (5 years). Secondly, this study incorporates data from hospitalization databases which helps capture all the IPD cases. Thirdly, this is a population-based study, representing BC's population of 4.8 million inhabitants, and which represents impact of PCV program from all locations, socio-economic statuses and comorbidities. Our study has several limitations. Firstly, in this retrospective cohort study, using laboratory surveillance data; there is a high likelihood of missed laboratory data in the initial years due to missed reporting or contamination of laboratory sample. To mitigate this effect, we have used concurrent hospitalization data to maximize case capture [15, 26] and we present main analyses from 2004 onwards, that is two years after the passive surveillance program was initiated and reporting was consistent. Secondly, we have conducted periodic trend analyses, which may underestimate or overestimate the overall impact based on the baseline burden of disease. However, to address this we have utilized all available data, and calculated rate ratios for specific age groups along with overall population. Thirdly, we were unable to assess the direct impact of the vaccination on occurrence of IPD, as we did not have complete data on vaccination status. In lieu of this missing data, general vaccination coverage trends from the provincial government helped us to understand vaccination uptake rates of PCV programs within our province [9].

## Conclusion

In conclusion, pneumococcal conjugate vaccines in the childhood immunization program have nearly eliminated invasive pneumococcal disease in children. Much of the decrease was seen after introduction of PCV7 but further reductions continued with the switch to PCV13, with gains in the immunized cohort of children and older adults. In addition, as there have been significant increases in non-vaccine serotypes, and continued surveillance of emerging pneumococcal disease will remain crucial to maintain effective immunization programs, and protect public health.

## Supporting information

**S1 Appendix.**
(DOCX)

## Acknowledgments

Foremost, I sincerely appreciate Dr. Fawziah Marra's valuable time and effort in providing constructive feedback and guidance in improving the scientific writing of this manuscript. I am grateful for all the support received from Tim Choi at Population Data BC and Peter Ng at British Columbia Center for Disease Control for acquisition and provision of administrative databases and laboratory surveillance data, respectively. Finally, I would like to thank my colleague, Ariana Saatchi for her valuable review and feedback for improving this manuscript.

## Author Contributions

**Conceptualization:** Nirma Khatri Vadlamudi, Fawziah Marra.

**Data curation:** Nirma Khatri Vadlamudi, Fawziah Marra.

**Formal analysis:** Nirma Khatri Vadlamudi.

**Funding acquisition:** Fawziah Marra.

**Investigation:** Nirma Khatri Vadlamudi.

**Methodology:** Nirma Khatri Vadlamudi.

**Project administration:** Nirma Khatri Vadlamudi, Fawziah Marra.

**Resources:** Nirma Khatri Vadlamudi, Fawziah Marra.

**Software:** Nirma Khatri Vadlamudi.

**Supervision:** David M. Patrick, Linda Hoang, Manish Sadarangani, Fawziah Marra.

**Validation:** Nirma Khatri Vadlamudi.

**Visualization:** Nirma Khatri Vadlamudi.

**Writing – original draft:** Nirma Khatri Vadlamudi.

**Writing – review & editing:** Nirma Khatri Vadlamudi, Fawziah Marra.

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
