## [Decision Letter · Decision Letter 0]

5 Jun 2020

PONE-D-20-08116

Incidence of invasive pneumococcal disease after introduction of the 13-valent conjugate pneumococcal vaccine in British Columbia: a retrospective cohort study

PLOS ONE

Dear Dr. Marra,

Thank you for submitting your manuscript to PLOS ONE. After careful consideration, we feel that it has merit but does not fully meet PLOS ONE’s publication criteria as it currently stands. Therefore, we invite you to submit a revised version of the manuscript that addresses the points raised during the review process.

We look forward to receiving your revised manuscript.

Kind regards,

Ka Chun Chong

Academic Editor

PLOS ONE

Journal Requirements:

2. In ethics statement in the manuscript and in the online submission form, please provide additional information about the patient records/samples used in your retrospective study. Specifically, please ensure that you have discussed whether all data/samples were fully anonymized before you accessed them and/or whether the IRB or ethics committee waived the requirement for informed consent. If patients provided informed written consent to have data/samples from their medical records used in research, please include this information.

4. Please include your tables as part of your main manuscript and remove the individual files. Please note that supplementary tables (should remain/ be uploaded) as separate "supporting information" files

Additional Editor Comments (if provided):

Reviewers' comments:

Reviewer's Responses to Questions

**Comments to the Author**

1. Is the manuscript technically sound, and do the data support the conclusions?

Reviewer #1: Yes

Reviewer #2: Yes

2. Has the statistical analysis been performed appropriately and rigorously? 

Reviewer #1: Yes

Reviewer #2: Yes

3. Have the authors made all data underlying the findings in their manuscript fully available?

Reviewer #1: Yes

Reviewer #2: Yes

4. Is the manuscript presented in an intelligible fashion and written in standard English?

Reviewer #1: Yes

Reviewer #2: Yes

5. Review Comments to the Author

Reviewer #1: The paper by Vadlamudi et al. describes a retrospective cohort study on the impact of pneumococcal vaccination, either with PCV7 or PCV13, on IPD in British Columbia during 2002-2015. The topic is of importance, but there are some issues that need to be dealt with. The first observation is that the results should be reported in a more comprehensive and organic way, not dividing them in primary and secondary analysis, jumping from the period of transition between PCV7 and PCV13 to the pre vaccination era and viceversa, both for IPD incidence and serotype analysis. It should be more easy for the reader to follow the overall IPD incidence and serotype trends during the entire study period and not going back and forth. As a consequence, Results proved to be quite confusing and should be rewritten avoiding too detailed descriptions of tables.

In addition, Authors should include a briefly description of the case definition and laboratory testing, with the definition of vaccine categories, in Methods and not as supplementary file. For clarity throughout the entire paper, it should be better to report PCV13 serotypes as PCV13-non PCV7 serotypes, and PPV23 serotypes as PPV23-non PCV13 serotypes.

In Discussion authors should comment more widely the increase observed in serotypes 19A and 7F following PCV13 use.

Minor revisions:

1. Throughout the entire paper it should be better to denominate the PCV7 and PCV13 periods as PCV7 era and PCV13 era, deleting the prefix ” post”.

2. Lines 79-80: specify to which years vaccination rate of 80% refers to.

3. Lines 155-157: description of serotype distribution should be deleted from this paragraph and included in a paragraph dedicated to serotype analysis.

4. It should be useful to include supplementary Figure 2 in the text.

Reviewer #2: This study reviews the results of a provincial-wide surveillance of IPD after the introduction of PCV7 and PCV13 into their children immunization programme during a 14-year period. The paper provides additional information about the effect of pneumococcal conjugate vaccines on IPD incidence and serotype distribution with similar results than those observed elsewhere.

6. PLOS authors have the option to publish the peer review history of their article (what does this mean?). If published, this will include your full peer review and any attached files.

Reviewer #1: No

Reviewer #2: No

---

## [Decision Letter · Decision Letter 1]

1 Sep 2020

PONE-D-20-08116R1

Incidence of invasive pneumococcal disease after introduction of the 13-valent conjugate pneumococcal vaccine in British Columbia: a retrospective cohort study

PLOS ONE

Dear Dr. Marra,

Thank you for submitting your manuscript to PLOS ONE. After careful consideration, we feel that it has merit but does not fully meet PLOS ONE’s publication criteria as it currently stands. Therefore, we invite you to submit a revised version of the manuscript that addresses the points raised during the review process.

We look forward to receiving your revised manuscript.

Kind regards,

Ka Chun Chong

Academic Editor

PLOS ONE

Additional Editor Comments (if provided):

Please address the remaining comments from the reviewer.

Reviewers' comments:

Reviewer's Responses to Questions

**Comments to the Author**

1. If the authors have adequately addressed your comments raised in a previous round of review and you feel that this manuscript is now acceptable for publication, you may indicate that here to bypass the “Comments to the Author” section, enter your conflict of interest statement in the “Confidential to Editor” section, and submit your "Accept" recommendation.

Reviewer #1: All comments have been addressed

Reviewer #2: All comments have been addressed

2. Is the manuscript technically sound, and do the data support the conclusions?

Reviewer #1: (No Response)

Reviewer #2: Yes

3. Has the statistical analysis been performed appropriately and rigorously? 

Reviewer #1: (No Response)

Reviewer #2: Yes

4. Have the authors made all data underlying the findings in their manuscript fully available?

Reviewer #1: (No Response)

Reviewer #2: Yes

5. Is the manuscript presented in an intelligible fashion and written in standard English?

Reviewer #1: (No Response)

Reviewer #2: Yes

6. Review Comments to the Author

Reviewer #1: This is a revised version of the manuscript “Incidence of invasive pneumococcal disease after introduction of the 13-valent conjugate pneumococcal vaccine in British Columbia: a retrospective color study ” by Vadlamudi et al. Authors revised parts of the manuscript according to some of the requests, and the manuscript appears quite improved, being the analysis of results more exaustive.

Some minor remarks:

1. For clarity, authors should move the sentence in lines 177-178 to line 172, before the last sentence of the first paragraph of Results.

2. Authors should correct the fields in Table 2.

3. Line 233: serotypes 15A, 15C, 23A, 23B, and 35F are not included in PPV23, therefore they should not be included in this paragraph.

4. Line 349: word “negated” seems to be too much imperative.

Reviewer #2: (No Response)

7. PLOS authors have the option to publish the peer review history of their article (what does this mean?). If published, this will include your full peer review and any attached files.

Reviewer #1: No

Reviewer #2: No

---

## [Decision Letter · Decision Letter 2]

15 Sep 2020

Incidence of invasive pneumococcal disease after introduction of the 13-valent conjugate pneumococcal vaccine in British Columbia: a retrospective cohort study

PONE-D-20-08116R2

Dear Dr. Marra,

We’re pleased to inform you that your manuscript has been judged scientifically suitable for publication and will be formally accepted for publication once it meets all outstanding technical requirements.

Kind regards,

Ka Chun Chong

Academic Editor

PLOS ONE

Additional Editor Comments (optional):

Reviewers' comments:

Reviewer's Responses to Questions

**Comments to the Author**

1. If the authors have adequately addressed your comments raised in a previous round of review and you feel that this manuscript is now acceptable for publication, you may indicate that here to bypass the “Comments to the Author” section, enter your conflict of interest statement in the “Confidential to Editor” section, and submit your "Accept" recommendation.

Reviewer #1: All comments have been addressed

2. Is the manuscript technically sound, and do the data support the conclusions?

Reviewer #1: Yes

3. Has the statistical analysis been performed appropriately and rigorously? 

Reviewer #1: Yes

4. Have the authors made all data underlying the findings in their manuscript fully available?

Reviewer #1: Yes

5. Is the manuscript presented in an intelligible fashion and written in standard English?

Reviewer #1: Yes

6. Review Comments to the Author

Reviewer #1: (No Response)

7. PLOS authors have the option to publish the peer review history of their article (what does this mean?). If published, this will include your full peer review and any attached files.

Reviewer #1: No

---

## [Editor Report · Acceptance letter]

17 Sep 2020

PONE-D-20-08116R2 

Incidence of invasive pneumococcal disease after introduction of the 13-valent conjugate pneumococcal vaccine in British Columbia: a retrospective cohort study 

Dear Dr. Marra:

I'm pleased to inform you that your manuscript has been deemed suitable for publication in PLOS ONE. Congratulations! Your manuscript is now with our production department. 

Kind regards, 

on behalf of

Dr. Ka Chun Chong 

Academic Editor

PLOS ONE